# SciNetBench: A Relation-Aware Benchmark for Scientific Literature Retrieval Agents

## Abstract

The rapid development of AI agent has spurred the development of advanced research tools, such as Deep Research. Achieving this require a nuanced understanding of the relations within scientific literature, surpasses the scope of keyword-based or embedding-based retrieval. Existing retrieval agents mainly focus on the content-level similarities and are unable to decode critical relational dynamics, such as identifying corroborating or conflicting studies or tracing technological lineages, all of which are essential for a comprehensive literature review. Consequently, this fundamental limitation often results in a fragmented knowledge structure, misleading sentiment interpretation, and inadequate modeling of collective scientific progress. To investigate relation-aware retrieval more deeply, we propose **SciNetBench**, the first **Sci**entific **Net**work Relation-aware **Bench**mark for literature retrieval agents. Constructed from a corpus of over 18 million AI papers, our benchmark systematically evaluates three levels of relations: ego-centric retrieval of papers with novel knowledge structures, pair-wise identification of scholarly relationships, and path-wise reconstruction of scientific evolutionary trajectories. Through extensive evaluation of three categories of retrieval agents, we find that their accuracy on relation-aware retrieval tasks often falls below 20%, revealing a core shortcoming of current retrieval paradigms. Notably, further experiments on the literature review tasks demonstrate that providing agents with relational ground truth leads to a substantial 23.4% performance improvement in the review quality, validating the critical importance of relation-aware retrieval. We publicly release our benchmark at https://anonymous.4open.science/r/SciNetBench/ to support future research on advanced retrieval systems.

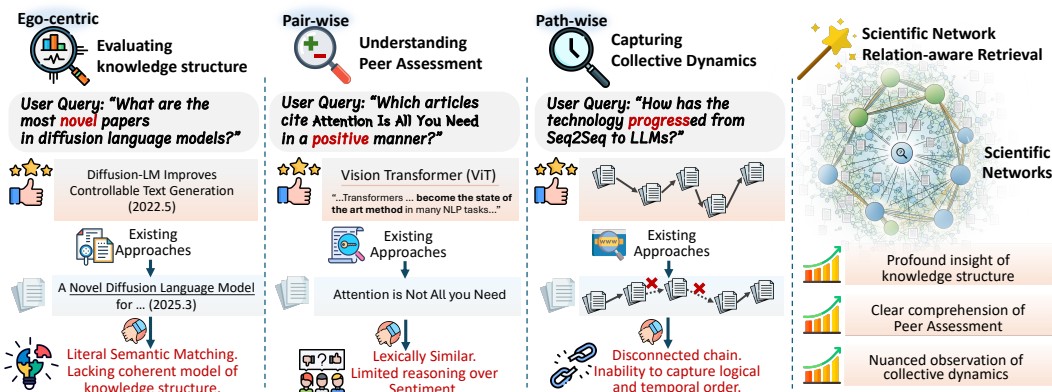

Figure 1: Importance of Scientific Networks in Literature Retrieval Scenarios

## 1 Introduction

The rapid development of AI agent has given rise to advanced research tools, such as ***Deep Research*** (OpenAI, 2025a), fostering the progress of automated scientific systems which are often referred to as AI Scientists (Yamada et al., 2025; Lu et al., 2024). The effective operation of such systems relies on a core capability: high-quality literature retrieval, which is essential to accurately

identify related work and position research projects within the existing literature. However, literature retrieval in scientific research is non-trivial, as it requires accurately understanding the deeper relations in the scientific network. Figure 1 illustrates three representative retrieval cases across different scholarly scenarios: *evaluating knowledge structure*, *understanding peer assessment*, and *capturing collective dynamics*. All these retrieval heavily rely on the scientific network, highlighting the critical importance of relation-aware retrieval.

However, existing retrieval agents generally struggle to achieve relation-aware retrieval, as also illustrated in Figure 1. Specifically, embedding-based agents (Beltagy et al., 2019; Cohan et al., 2020; Huang et al., 2020) rely solely on static representations of the literature, limiting them to shallow semantic matching and preventing multi-step, agentic reasoning. Meanwhile, Deep Research agents (He et al., 2025; Lála et al., 2023; Skarlinski et al., 2024), despite their iterative pipelines, cannot explicitly model and fully leverage the relations encoded in the scientific network. Furthermore, existing literature retrieval benchmarks also overlooked the deep relations in the scientific networks. Most benchmarks focus primarily on semantic precision, evaluating whether retrieved results are relevant in terms of domain or topic (Ajith et al., 2024; He et al., 2025). Although STARK introduces the notion of network (Wu et al., 2024), it remains limited to structured hops between entities such as authors, sections, and papers, without addressing the deeper relations that exist directly between the publications themselves.

Motivated by this, we propose **SciNetBench**, the first **Sci**entific **Net**work Relation-aware **Bench**mark to systematically evaluate how well literature retrieval agents capture these relations within the scientific network. Owing to the need of open-access publications and the fact that agent-based tools are most prevalent in the discipline of AI, we concentrate our benchmark on AI. Constructed from a corpus of 18,639,141 AI papers, SciNetBench cover all major AI subfields. To capture different levels of relational structures, we design three categories of tasks:

- **Ego-centric**: Retrieving papers based on individual intrinsic scientific properties, such as identifying the most novel or disruptive work within a research field.
- **Pair-wise**: Identifying the specific relational context between two papers, such as whether one supports, contradicts, or builds upon the other.
- **Path-wise**: Retrieving citation paths that capture the evolution of scientific ideas, such as reconstructing the development trajectory from a foundational concept to a SOTA method.

We conduct extensive evaluations of eight retrieval methods across three categories on our benchmark: (1) retrieval via embedding models, (2) retrieval via agentic models, (3) retrieval via deep research pipelines. Experimental results demonstrate that current approaches consistently fail across all three proposed tasks. We further evaluate the impact on downstream applications through *literature review*, demonstrating that the lack of relation-aware retrieval results in incomplete or inaccurate literature summaries and highlighting the crucial importance of efficient exploitation of the literature network. Our contributions can be summarized as follows:

- We propose the **first benchmark** for relational retrieval in scientific literature, constructed from a corpus of over 18 million AI papers. It provides standardized tasks, queries, and evaluation metrics to assess capabilities beyond semantic matching.
- We introduce a **novel taxonomy** of scientific relational retrieval through three granularities: ego-centric, pair-wise, and path-wise. This taxonomy provides a conceptual framework for understanding scholarly relationships.
- Through extensive evaluation of eight retrieval methods, we **quantify the limitations** of these methods and demonstrate the critical importance of relational retrieval. Additional experiments further validate the benefits of literature network for downstream applications.

## 2 BENCHMARK OVERVIEW AND EVALUATED MODELS

### 2.1 BENCHMARK OVERVIEW

*Scientific network* refers to a semantically enriched graph of scholarly publications, constructed from citation relations among papers and citation contexts within documents, so as to capture not only the presence of citation links but also their underlying semantic nature. To build such a network, we first leverage OpenAlex (RELEASE 2025-07-07), a comprehensive open scholarly dataset, to obtain

citation relations among scientific papers. We downloaded the complete data snapshot directly from its official Amazon S3 bucket* , which contains metadata for 269,091,010 papers, including titles, abstracts, authorship, citations, and publication information. To construct an AI-specific corpus, we identified 177 AI-related subtopics (e.g., three-dimensional reconstruction, tool-augmented reasoning) using OpenAlex's topic classification system through manual review and aggregation (see A.2 for the full list). This filtering resulted in a candidate pool of 18,639,140 AI papers, which to our knowledge represents the largest curated literature base for retrieval benchmarking to date. Additionally, we incorporated the complete arXiv PDF corpus (as of July 7, 2025) for auxiliary text extraction and validation.

From this corpus, we constructed 1,087 high-quality queries across three relational tasks: 354 (32.6%) for ego-centric retrieval, 600 (55.2%) for pair-wise relation identification, and 133 (12.2%) for path-wise evolutionary analysis. Queries were first generated through structured rules (e.g., leveraging citation and topics) and subsequently validated and refined through manual expert review, ensuring both coverage and reliability. The distribution across tasks reflects their relative prevalence and importance in real-world scientific reasoning, with pair-wise relations being most common in scholarly discourse and path-wise trajectories representing more complex, higher-level reasoning. Our benchmark contains no private or non-public information.

## 2.2 Evaluated Models

We evaluate a total of eight retrieval models across three categories:

**Category I: Retrieval via Embedding Models: SciBERT** (Beltagy et al., 2019) is the first embedding model specifically trained on scientific literature. It was pre-trained on a corpus of 3.17 billion tokens, predominantly from the biomedical domain. It also constructed a new WordPiece vocabulary directly from the scientific corpus, which significantly enhanced the model's representational capacity. More recently, with the rapid advances in large language models (LLMs), their embedding layers have also been regarded as reliable embedding models. In our benchmark, we include the newly released and powerful **Qwen3-8B-Embedding** (Zhang et al., 2025) model in evaluation.

**Category II: Retrieval via Agentic Models:** This category encompasses frameworks that employ agentic workflows for information retrieval and synthesis. We include **paperQA2** (Skarlinski et al., 2024), a recently released system by FutureHouse designed for high-accuracy, retrieval-augmented QA over scientific documents. Its agent-driven framework integrates vector retrieval with LLM-based comprehension, first segmenting the corpus into discrete text chunks and indexing them individually, then executing a pipeline that includes evidence gathering and answer generation with explicit citation support. Also selected is **PaSa** (He et al., 2025), which operates through a Crawler and a Selector. The Crawler autonomously generates search queries from user input, retrieves relevant papers, and iteratively expands the search scope through citation tracking. The Selector then evaluates the relevance of the retrieved papers to the query. Further included are **gpt-4o-mini-search** and **gpt-4o-search** (OpenAI, 2025b), which leverage LLMs to perform multi-step reasoning. These models are equipped with powerful web search tools, enabling them to autonomously generate queries, retrieve information from diverse online sources, and synthesize responses.

**Category III: Retrieval via Deep Research Agents:** We selected **o4-mini-deep-research** and **o3-deep-research** (OpenAI, 2025a). Deep Research agents operate through a deeply iterative pipeline, in which they autonomously decompose complex queries into sub-tasks and dynamically adapt retrieval strategies. They are capable of executing dozens of iterative search-and-reasoning cycles before the final answer.

## 3 Ego-centric Relation: Evaluating Knowledge Structures through Scientometric Insights

### 3.1 Evaluation Protocol

**Construction:** Beyond merely finding related papers, deep scientific inquiry often requires evaluating the intrinsic scholarly value of individual publications, such as identifying truly novel or disruptive work. To address this need, we introduce Ego-centric Retrieval, a category focused on

---

*https://docs.openalex.org/download-all-data/download-to-your-machine

assessing papers based on their knowledge structure. Literature attributes like novelty often arise from distinctive configurations in a paper's knowledge structure, such as the pioneering combination of concepts from previously disconnected fields. By leveraging scientometric indicators, we can quantifies such structural characteristics to answer queries like, "Which is the most novel paper in diffusion language models?", a task beyond semantic matching, as it requires comprehension of abstract, structure-derived properties.

To perform this task, we make use of the collective knowledge embedded in citation networks. For example, a paper's citation patterns, how it is cited by later work, can serve as a reliable indicator of its novelty, and disruptiveness. Building on this idea, we convert two established scientometrics indicators, the *novelty* and the *disruption index*, into concrete retrieval queries. This allows us to evaluate whether current retrieval models can correctly interpret and respond to queries aimed at capturing the deeper scientific value of papers.

Specifically, we draw on the method proposed by Uzzi et al. (Uzzi et al., 2013) for measuring **novelty**: by analyzing co-citation pairs within a paper's references, they quantify the extent to which the work combines rare or "atypical" knowledge components. This is calculated by converting each co-citation pair's frequency into a Z-score relative to the disciplinary norm, with the paper's final novelty score being the 10th percentile ($p_{10,z}$) of these scores. A lower score thus signifies a more novel combination of knowledge. In parallel, we adopt the **disruption index** introduced by Funk and Owen-Smith (Funk & Owen-Smith, 2017). This metric evaluates whether subsequent publications continue to cite both the focal paper and its predecessors, or instead shift to citing only the focal paper. The index is calculated as $(N_i - N_j)/(N_i + N_j)$, where $N_i$ is the count of papers citing only the focal work and $N_j$ is the count citing both the focal work and its references, thereby characterizing whether the work extends existing trajectories or fundamentally disrupts prior research.

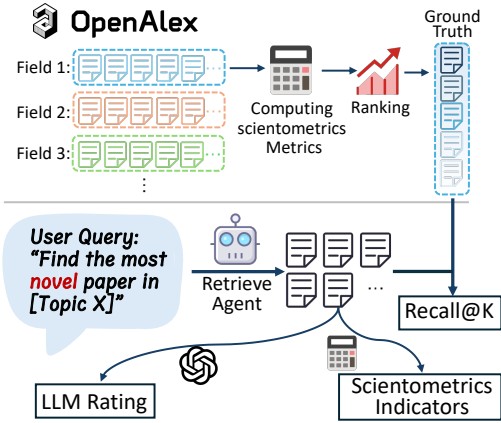

Figure 2: Evaluation Protocol of Ego-Centric Retrieval.

**Evaluation:** For evaluation, each query is used to retrieve a ranked list of 50 candidate papers. We employ a multi-faceted assessment strategy covering three distinct aspects. First, for our scientometrics indicators, we calculate the raw novelty and disruption scores for each retrieved paper. The raw novelty score is a Z-score that is theoretically unbounded, where more negative values indicate higher novelty, while the disruption index ranges from -1 (consolidating) to +1 (disruptive). Given the distinct and unintuitive scales of these raw scores, we convert each into a percentile rank against a global reference corpus of millions of papers. The average of these percentile ranks for the candidates constitutes our final *Novelty-SoS* and *Disruption-SoS* metrics.

Second, we incorporate LLMs (GPT-5) to provide complementary semantic judgments, yielding the *Novelty-LLM* and *Disruption-LLM* metrics. These models assign a score from 1 to 10 for each concept based solely on the paper's title and abstract. Third, we establish ground truth labels to measure retrieval performance. For each of the 177 AI subfields, we identify the top 50 most novel and top 50 most disruptive papers based on their scientometric ranks, creating two distinct ground truth sets. Performance is then measured using *Novelty-Recall@50* and *Disruption-Recall@50*, which evaluate the system's ability to include these key papers within its top 50 results.

### 3.2 EXPERIMENTAL RESULTS

As shown in Table 1, a clear performance hierarchy is evident across all evaluation metrics for ego-centric retrieval. Deep Research systems (e.g., o3-deep-research) consistently achieved the best results, followed by web search-based agentic models (e.g., gpt-4o-search), while other models demonstrated substantially weaker performance. This demonstrates that agentic workflows and flexible use of web tools can effectively enhance performance in literature retrieval. Nevertheless, even the top-performing systems struggled significantly according to recall-based evaluation: the best

| Category | Models | Novelty-SoS | Novelty-LLM | Novelty-Recall@50 | Disruption-SoS | Disruption-LLM | Disruption-Recall@50 |
|---|---|---|---|---|---|---|---|
| **Embedding** | SciBERT | 3.686 | 2.382 | 0.00% | 4.299 | 2.201 | 0.00% |
| | Qwen3-Embed | 4.115 | 2.836 | 0.20% | 3.658 | 3.045 | 2.25% |
| **Agentic** | PaSa | 4.646 | 5.059 | 0.10% | 6.050 | 3.831 | 2.68% |
| | PaperQA | 5.512 | 5.491 | 0.17% | 5.675 | 3.970 | 2.44% |
| | gpt-4o-mini-search | 6.352 | 6.196 | 0.60% | 6.656 | 5.874 | 3.12% |
| | gpt-4o-search | 6.447 | 6.275 | 1.10% | 6.711 | 5.968 | 4.35% |
| **DeepResearch** | o4-mini-deep-research | 6.815 | 6.490 | 1.30% | 6.910 | **7.060** | **4.60%** |
| | o3-deep-research | **6.951** | **6.491** | **1.42%** | **6.956** | 6.970 | 3.92% |

Table 1: Performance for Frontier Models and Agents on Ego-Centric Tasks

recall@50 for novelty was only 1.42%, and for disruption only 4.60%, indicating that over 95% of truly groundbreaking papers were missed by all systems.

This pattern was consistent across both scientometrics scores and LLM-based assessment, revealing that current retrieval approaches are misaligned with the demands of scientific practice, where accurate assessment of papers based on their intrinsic scientific properties is essential. The observed failures underscore the necessity of developing relation-aware retrieval models capable of understanding scholarly networks and capturing the deeper scientific value of papers.

# 4 Pair-wise Relation: Understanding Peer Assessment through Citation Contexts

## 4.1 Evaluation Protocol

Building upon the scientometrics indicators discussed previously, which primarily focus on the statistical properties of individual nodes within the scholarly network, this section extends the analysis to pairwise relations between papers, with particular emphasis on fine-grained semantic associations derived from citation contexts (peer assessment). Accurately identifying such relational information is critical for multiple downstream scientific applications: it enables high-precision literature recommendation by moving beyond topical similarity to capture nuanced scholarly dialogues; it significantly improves the quality of retrieval-augmented generation (RAG) systems by providing evidence chains with explicit sentiment and contextual labels; and it supports the construction of richly structured knowledge graphs that reflect the true discursive landscape of a field, facilitating advanced analyses such as trend detection, controversy mapping, and knowledge gap identification.

To operationalize this focus, we designed a suite of pairwise retrieval tasks encompassing two critical types of scholarly relationships. The first type involves **sentiment-oriented queries**, such as "Which papers cite Paper XX positively?", requiring systems to distinguish between critical, supportive, or neutral citations based on contextual sentiment. The second type targets **context-based co-mention queries**, exemplified by "Which papers are frequently mentioned together with Paper XX within the same paragraph?". This task demands the identification of papers jointly referenced within a coherent narrative segment (e.g., a paragraph in the related work section), thereby capturing methodological comparisons, or thematic contrasts within the scientific literature.

**Evaluation:** Retrieval quality is assessed through four complementary metrics. ***Cite-Acc*** measures whether a retrieved paper is actually cited by the query paper. ***Cite-Sentiment*** extends the evaluation by analyzing the sentiment of the citation: each retrieved paper's PDF is obtained from arXiv and parsed with *GROBID* Lopez & Romary (2013), which provides both the reference list and in-text citation links; verified citations are then traced to their surrounding paragraphs, where GPT-5 determines whether the citation is positive, negative, or neutral. ***CoMention-Acc*** captures contextual co-citation by checking in the citation network whether a source article cites both the retrieved paper and the query paper. Building on this, ***CoMention-Paragraph*** requires stronger evidence by parsing the co-citing article with *GROBID* to confirm that both citations not only appear but also co-occur within the same paragraph, thereby ensuring that co-citation evidence is grounded in explicit textual context rather than inferred solely from the network.

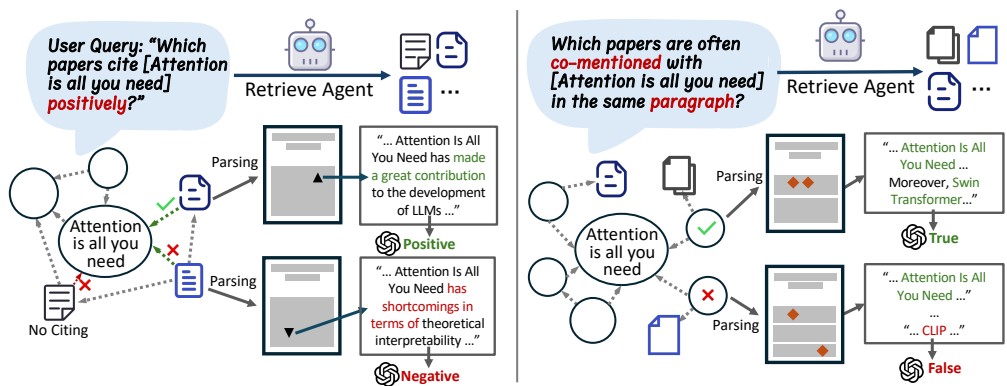

Figure 3: Evaluation Protocol of Pair-Wise Retrieval.

| Category | Models | Cite-Acc | Cite-Sentiment | CoMention-Acc | CoMention-Paragraph |
|---|---|---|---|---|---|
| **Embedding** | SciBERT | 1.80% | 0.00% | 0.00% | 0.00% |
| | Qwen3-Embed | 10.65% | 3.07% | 23.95% | 5.32% |
| **Agentic** | PaSa | 18.31% | 4.89% | 26.31% | 7.63% |
| | PaperQA | 8.20% | 3.45% | 8.25% | 7.11% |
| | gpt-4o-mini-search | 46.17% | 12.28% | 66.04% | 6.89% |
| | gpt-4o-search | 46.36% | 9.28% | 68.82% | 7.32% |
| **DeepResearch** | o4-mini-deep-research | 61.89% | **16.77%** | 68.60% | 10.10% |
| | o3-deep-research | **62.84%** | 13.80% | **76.88%** | **16.30%** |

Table 2: Performance for Frontier Models and Agents on Pair-Wise Tasks

## 4.2 EXPERIMENTAL RESULTS

Table 2 presents the performance of different systems on the pair-wise tasks. The results indicate that current agentic retrieval approaches, including both web search tools and reasoning-based agents, provide improvements in retrieval quality, with deep research platforms yielding even more substantial gains. For example, *Cite-Acc* reaches around 46% for agentic models and exceeds 60% for deep research systems, while *CoMention-Acc* can be as high as 77%. These findings suggest that leveraging external search capabilities or reasoning mechanisms enables models to more effectively identify citation links and co-citation patterns compared to purely embedding-based methods.

However, substantial challenges remain. Metrics such as *Cite-Sentiment* and *CoMention-Paragraph* continue to show low performance across all approaches, indicating that capturing citation sentiment and grounding co-mentioned papers within the same paragraph remain difficult. Overall, while advanced retrieval methods enhance citation detection, relational and context-aware reasoning is still far from solved, clearly highlighting the necessity of leveraging the literature network for document retrieval.

## 5 PATH-WISE RELATION: CAPTURING COLLECTIVE DYNAMICS OF SCIENTIFIC EVOLUTION

### 5.1 EVALUATION PROTOCOL

**Construction:** The previously introduced ego-centric and pair-wise tasks assess models' ability to capture intrinsic properties and binary relations. However, scientific progress typically unfolds as an evolving narrative, where new ideas build upon prior work in multi-step trajectories, forming the collective dynamics of scientific knowledge. To capture this essential aspect, we propose our third category: *Path-Wise Retrieval*, which evaluates whether a system can reconstruct the evolutionary path connecting a sequence of papers. For example, a researcher might ask: "What are the key milestones linking the seminal Transformer paper to today's large language models?" Answering such queries requires understanding not just paper relevance, but also the logical and citational dependencies that form a coherent developmental chain.

The importance of this task lies in its centrality to literature reviews and research trend analysis. A system that merely outputs unordered related papers cannot reveal the intellectual structure of a field. Yet, existing retrieval methods are almost entirely incapable of constructing such paths, as they lack mechanisms for modeling temporal progression or causal reasoning in scholarly lineage. By introducing the path-wise task, our benchmark offers the first rigorous testbed for evaluating retrieval systems on their ability to reconstruct scientific evolution, pushing them beyond shallow retrieval toward genuine knowledge synthesis.

To construct meaningful technological evolution queries, we leveraged the aforementioned 177 AI subfields. For each subfield, we retrieved the top 50 most-cited papers of all time (treated as classical papers) and the top 10 most-cited papers since 2024 (treated as emerging papers). By randomly pairing classical and emerging papers, we generated a large set of candidate pairs. We then applied the OpenAlex citation network to filter out paper pairs that are topologically connected, followed by manual inspection to ensure that the paired papers remain thematically coherent. This yielded a total of 133 high-quality queries, such as: "What is the most influential citation path from *Attention Is All You Need* to *An Image is Worth 16x16 Words: Transformers for Image Recognition at Scale*?"

**Evaluation:** For each query, we constructed the ground truth path using a breadth-first search (BFS) algorithm. Specifically, we enumerated all connecting paths between the two endpoint papers and computed the cumulative citation count of all papers along each path. The path with the highest total citation count was selected as the ground truth. This approach is well justified, as it closely parallels the notion of collective credit allocation proposed by Hua-Wei Shen et al. (Shen & Barabási, 2014), where

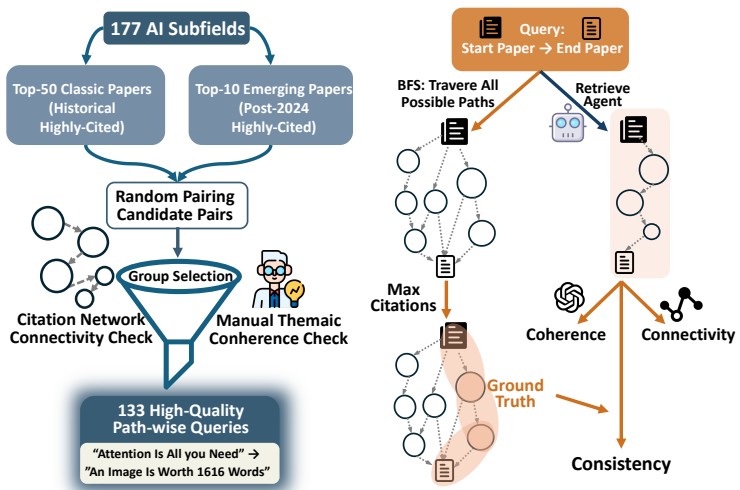

Figure 4: Evaluation Protocol of Path-Wise Retrieval.

citations are interpreted as community votes that represent collective recognition of a research trajectory. We evaluated retrieval results along three complementary dimensions. First, ***Consistency*** measures the degree of overlap between the retrieved path and the ground-truth path. Second, ***Connectivity*** evaluates whether the retrieved papers form a connected citation path linking the query endpoints within the citation network. Third, ***Rationality*** measures the plausibility of the retrieved path. We first evaluate this using an LLM, which is prompted with the titles and abstracts of the retrieved papers to judge whether the sequence forms a coherent and reasonable evolutionary narrative.To further enhance reliability, we complement this with a human evaluation: for a subset of 50 representative queries, three AI-expert annotators independently scored the retrieved paths from each model on a 1—10 scale based on rationality. The scores are then averaged to form a *Rationality-Human* metric, which is included as an additional column in the table.

## 5.2 EXPERIMENTAL RESULTS

Results shown in Table 3 also reveal a pronounced performance gap between traditional embedding-based methods and more advanced retrieval paradigms on the path-wise task. Embedding models such as SciBERT and Qwen3-Embed essentially fail, achieving near-zero *Consistency* and *Connectivity*, and very low scores in both LLM-judged (*Rationality-LLM*) and human-judged (*Rationality-Human*) evaluations. This indicates that these models cannot capture sequential dependencies or reconstruct coherent scientific trajectories beyond surface-level semantic similarity.

In contrast, web search and deep research systems demonstrate substantially stronger performance. Models such as *gpt-4o-search* and *o3-deep-research* achieve over 60% *Consistency* and significantly higher *Connectivity*, successfully retrieving papers that lie along true evolutionary paths. Deep re-

| Category | Models | Consistency | Connectivity | Rationality-LLM | Rationality-Human |
|---|---|---|---|---|---|
| **Embedding** | SciBERT | 0% | 0% | 1.195 | 1.60 |
| | Qwen3-Embed | 2.11% | 3.00% | 3.024 | 2.25 |
| **Agentic** | PaSa | 4.30% | 2.1% | 2.405 | 2.50 |
| | PaperQA | 5.56% | 2.50% | 1.988 | 2.14 |
| | gpt-4o-mini-search | 51.76% | 7.23% | 4.451 | 4.98 |
| | gpt-4o-search | **65.45%** | 10.24% | 4.651 | 5.25 |
| **DeepResearch** | o4-mini-deep-research | 61.33% | 12.00% | 6.700 | 6.65 |
| | o3-deep-research | 63.10% | **14.00%** | **6.840** | **7.04** |

Table 3: Performance for Frontier Models and Agents on Path-Wise Tasks

search models also outperform others in *Rationality*, with both LLM and human evaluations confirming that the retrieved sequences form more coherent and plausible narratives of scientific progress. These results highlight that reconstructing intellectual lineages requires relation-aware retrieval, and they validate the path-wise task as a rigorous benchmark for assessing advanced retrieval capabilities that go beyond surface-level semantic matching.

# 6 DEMONSTRATING REAL-WORLD VALUE OF SCINETBENCH VIA DOWNSTREAM APPLICATIONS

To demonstrate the practical value of our benchmark, we conducted a case study on **automatic literature review**. Through this experiment, we aim to illustrate, from an application perspective, the critical importance of relation-aware retrieval.

**Experimental Setup:** We selected a set of 20 representative queries from our path-wise benchmark, each corresponding to a research path with ground-truth papers and abstracts. For each query, the ground-truth sequence provides a reference trajectory of the evolution of ideas, allowing for systematic evaluation of survey generation methods.

Four different approaches were evaluated: *Base LLM*: A LLM that generates surveys solely from the input query and paper abstracts, without any external retrieval or ground-truth information. *Search-enabled LLM*: LLM leverages web-based literature retrieval tools to generate survey reports without access to ground-truth paper sequences. *Deep Research System*: Our proposed system, which explicitly models literature evolution and performs multi-hop retrieval over relational structures. *Base LLM with Ground Truth*: The same model as above, but provided with the ground-truth paper sequences for

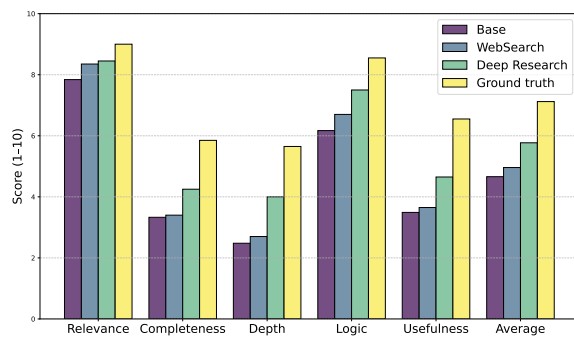

Figure 5: LLM evaluation of survey generation quality.

each query to assess the upper-bound performance achievable when the full evolution path is known. All methods used identical prompt templates, emphasizing structured survey writing in academic Markdown style, highlighting the progression and connections between papers, and focusing on relevance, completeness, depth, logical flow, and overall usefulness.

**Evaluation Metrics:** To quantify survey quality, we adopted two complementary protocols: (1) *LLM Automatic scoring*: Each generated report was evaluated along five dimension: *Relevance*, *Completeness*, *Depth*, *Logical Consistency*, and *Usefulness*. Scores were assigned on a scale of 1–10, and aggregated averages across queries were computed for each method (Figure 5). (2) *Human preference ranking*: Three domain experts were asked to comparatively rank the four systems' outputs for each query (from most to least useful). Table 4 summarizes the distribution of ranks and average scores.

**Results and Analysis:** Across both automatic evaluation metrics (Figure 5) and human preference rankings (Table 4), the ground-truth-assisted model attains the highest scores on every evaluated dimension, confirming the clear upper bound when accurate research trajectories are available. Deep-Research system is the strongest baseline: it substantially outperforms the rest of the models, most notably in Completeness (4.25) and Depth (4.00). By contrast, the Base LLM and Search-enabled LLM lag behind; the search-enabled model attains comparatively high relevance but exhibits low completeness and depth, while the base model shows only modest gains in logical coherence. The concordance between automatic evaluation and human rankings indicates that relation-aware retrieval materially improves survey quality, yet the remaining gap in completeness and depth between Deep Research and Ground Truth highlights the need for better modeling of literature relations.

Taken together, these findings underscore that **capturing literature relations is critical for practical downstream applications**. Systems with access to richer relational context produce more coherent, informative, and useful survey reports. Beyond literature review, literature relations can also provide significant benefits

| Method | Avg. Rank ↓ | #Rank=1 | #Rank=2 | #Rank=3 | #Rank=4 |
|---|---|---|---|---|---|
| Ground Truth | **1.25** | 15 | 4 | 1 | 0 |
| Deep Research System | 2.00 | 3 | 12 | 4 | 1 |
| Base LLM | 3.10 | 2 | 3 | 10 | 5 |
| Search-enabled LLM | 3.65 | 0 | 1 | 5 | 14 |

Table 4: Human preference rankings across 20 queries. Lower average rank indicates better overall preference.

in other applications such as automated experiment design, innovation ideation, and scientific knowledge discovery. This highlights the practica value of our benchmark in supporting the development of systems capable of nuanced scientific reasoning that directly benefits real-world applications.

# 7 RELATED WORKS

Several benchmarks have been proposed to advance scientific literature retrieval. Here, we review representative efforts and highlight how our work emphasizes relational understanding beyond thematic or entity-centric retrieval.

LitSearch (Ajith et al., 2024) constructs queries using two complementary strategies. *Inline-citation questions* sample paragraphs with citations from research papers, then GPT-4 rewrites them into literature search questions answerable by the cited works. *Author-written questions* are crafted by paper authors, guided by realism, specificity, and resistance to trivial keyword-based resolution.

The PASA (He et al., 2025) benchmark similarly generates queries from *related work* sections using LLMs (e.g., GPT-4o) and expands candidate sets via conventional and academic search engines, search-augmented ChatGPT, and LLM rewriting, with manual expert filtering. Both PASA and LitSearch primarily focus on topical localization, identifying papers in specific domains. In contrast, our benchmark emphasizes reasoning over scholarly relationships, such as methodological influence, disruptive contributions, and conceptual development. STARK (Wu et al., 2024) builds a semi-structured database from the Microsoft Academic Graph, supporting structured knowledge queries that require multi-hop reasoning over predefined entities. Unlike STARK, our benchmark evaluates the discovery of implicit, semantically rich connections among scientific works, providing a more natural testbed for deep scientific reasoning.

# 8 CONCLUSION

In this paper, we propose a benchmark, **SciNetBench**, for relation-aware retrieval in scientific literature. SciNetBench evaluates retrieval systems across three granularities: **ego-centric** tasks that focus on individual papers' intrinsic scientific properties, **pair-wise** tasks that assess the relationships between two papers, and **path-wise** tasks that reconstruct citation paths to capture the evolution of scientific ideas. Our experiments demonstrate that current retrieval methods struggle to capture these relational structures, and that this deficiency can substantially degrade downstream applications such as literature review. By emphasizing relational understanding over isolated texts or semantic similarity alone, SciNetBench highlights the practical necessity of integrating scholarly relations, providing a foundation for developing retrieval systems capable of nuanced scientific reasoning and more reliable knowledge synthesis.

## REPRODUCIBILITY STATEMENT

We provide all resources and code necessary to use our SciNetBench. All datasets and models employed are fully open-source or publicly accessible, and no privacy or copyright concerns are involved. All datasets and models are cited in Section 2.2. Our project code and dataset, including the implementation of evaluation scripts, queries, and metrics, are publicly available via the following anonymous link: `https://anonymous.4open.science/r/SciNetBench/`.

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

# A APPENDIX

## A.1 EXPERIMENTAL DETAILS

For both SciBERT and Qwen3-Embedding Model, we processed a total of 18,639,140 papers individually, concatenating each paper's title and abstract into a single string in the format "title: <title>, abstract: <abstract>". We then obtained embeddings for each concatenated string using the respective embedding model. The embedding dimensionality is 768 for SciBERT and 4096 for Qwen3-Embed.

To enable efficient large-scale similarity search over these high-dimensional embeddings, we constructed a dedicated index using the Faiss library. The preprocessing pipeline first performs L2 normalization on all embeddings, which ensures that maximum inner-product search is equivalent to finding the highest cosine similarity, a standard metric for semantic relevance. We employ an IndexIVFPQ structure, which combines an inverted file system (IVF) for coarse partitioning of the search space and product quantization (PQ) for compact vector representation. Specifically, the algorithm first partitions the entire vector space into 16,384 cells using k-means clustering, where each cell is represented by a centroid vector. This IVF structure enables a substantial pruning of the search space by only examining a small subset of cells closest to the query vector. Within each cell, vectors are further compressed using product quantization: each vector is split into 32 sub-vectors, and each sub-vector is quantized into an 8-bit code pointing to the nearest centroid in a learned codebook.

This two-level scheme, coarse quantization via IVF followed by fine-grained PQ, significantly reduces memory footprint while accelerating retrieval, achieving a favorable trade-off between search accuracy and efficiency. All embeddings and queries are computed on a single NVIDIA A100 GPU.

For PASA and PaperQA2, we strictly followed the implementations provided in their respective GitHub repositories. PASA was deployed on a local A100 GPU using its pretrained pasa-7b-crawler and pasa-7b-selector models, with API keys configured for Google Search and other relevant tools. For PaperQA2, we performed segmentation and embedding on all papers to fully leverage the model's capabilities. For all OpenAI models, we accessed them using official API keys from the OpenAI platform.

## A.2 AI RESEARCH FIELDS AND SUBFIELDS

- **Reinforcement Learning**
  - Model-Free Reinforcement Learning
  - Model-Based Reinforcement Learning
  - Offline Reinforcement Learning
  - Hierarchical Reinforcement Learning
  - Inverse Reinforcement Learning
  - Safe Reinforcement Learning
  - Multi-Agent Reinforcement Learning
  - Reinforcement Learning with Function Approximation
  - Reinforcement Learning for Control and Robotics

- **Bandits and Multi-Armed Bandits**
  - Stochastic Bandits
  - Contextual Bandits
  - Linear Bandits
  - Combinatorial Bandits
  - Bandits with Knapsacks
  - Pure Exploration and Best-Arm Identification

- **Imitation Learning**
  - Behavior Cloning
  - Inverse Reinforcement Learning
  - Generative Adversarial Imitation Learning
  - Dataset Aggregation
  - Offline Imitation Learning

- **Self-Supervised Learning**
  - Contrastive Learning
  - Masked Autoencoders
  - Predictive Coding
  - Bootstrap Your Own Latents
  - Clustering-based Self-Supervised Learning

- **Contrastive Learning**
  - Information Noise Contrastive Estimation Loss and Mutual Information Estimation
  - Supervised Contrastive Learning
  - Hard Negative Mining
  - Cross-Modal Contrastive Learning
- **Federated Learning**
  - Federated Averaging
  - Personalized Federated Learning
  - Privacy-Preserving Federated Learning
  - Communication-Efficient Federated Learning
  - Federated Learning with Heterogeneous or Non-IID Data
- **Continual Learning and Lifelong Learning**
  - Regularization-based Methods
  - Replay-based Methods
  - Parameter Isolation Methods
  - Task-Free and Online Continual Learning
  - Concept Drift Adaptation
- **Online Learning**
  - Online Gradient Descent
  - Online Convex Optimization
  - Online-to-Batch Conversion
  - Adaptive Learning Rates
- **Multi-Task Learning**
  - Hard Parameter Sharing
  - Soft Parameter Sharing
  - Task Relation Learning
  - Multi-Objective Optimization
- **Active Learning**
  - Uncertainty Sampling
  - Query by Committee
  - Expected Model Change
  - Bayesian Active Learning
- **Transfer Learning**
  - Feature-based Transfer
  - Fine-Tuning Pretrained Models
  - Domain Adaptation
  - Multi-Domain Training
- **Domain Adaptation and Generalization**
  - Unsupervised Domain Adaptation
  - Semi-Supervised Domain Adaptation
  - Domain Generalization
  - Adversarial Domain Adaptation
- **Meta-Learning**
  - Gradient-based Meta-Learning
  - Metric-based Meta-Learning
  - Black-box Meta-Learning
  - Meta-Optimization
- **Human-in-the-Loop and Human Feedback**
  - Interactive Labeling
  - Preference Learning
  - Human-Guided Model Editing
- **Reinforcement Learning from Human Feedback**
  - Reward Model Training
  - Preference Elicitation
  - Policy Optimization with Human Feedback
  - Direct Preference Optimization
- **Generative Models**
  - Autoregressive Models
  - Normalizing Flows
  - Energy-Based Models
  - Score-Based Models
- **Diffusion Models**
  - Denoising Diffusion Probabilistic Models
  - Latent Diffusion Models
  - Score Matching and Stochastic Differential Equation Approaches
  - Guided Diffusion for Text-to-Image Generation
- **Graph Neural Networks**
  - Graph Convolutional Networks
  - Graph Attention Networks
  - Graph Isomorphism Networks
  - Spatio-Temporal Graph Neural Networks
  - Heterogeneous Graph Learning
- **Transformers**

- – Encoder-Only Transformers
- – Decoder-Only Transformers
- – Encoder-Decoder Transformers
- – Long-Context Transformers
- – Efficient Transformers

- **Deep Learning Theory**
  - – Generalization Bounds
  - – Double Descent Phenomenon
  - – Neural Tangent Kernel Theory
  - – Implicit Bias in Gradient Descent

- **Natural Language Processing**
  - – Language Modeling
  - – Text Classification
  - – Named Entity Recognition
  - – Question Answering
  - – Information Extraction
  - – Machine Translation
  - – Text Summarization
  - – Dialogue Systems

- **Large Language Models**
  - – Pretraining Objectives
  - – Scaling Laws
  - – Fine-Tuning Methods
  - – Alignment Methods
  - – Evaluation and Benchmarking of Language Models

- **Large Language Model Agents**
  - – Tool-Augmented Reasoning
  - – Planning and Task Decomposition
  - – Memory-Augmented Agents
  - – Function Calling and API Orchestration

- **Prompt Engineering and In-Context Learning**
  - – Zero-Shot Prompting
  - – Few-Shot Prompting
  - – Chain-of-Thought Reasoning
  - – Automatic Prompt Search

- **Computer Vision**
  - – Image Classification
  - – Object Detection
  - – Semantic and Instance Segmentation
  - – Pose Estimation
  - – Three-Dimensional Reconstruction

- **Vision-Language Models**
  - – Contrastive Pretraining
  - – Image Captioning
  - – Visual Question Answering
  - – Grounded Language Understanding

- **Time Series Analysis and Forecasting**
  - – Autoregressive Models for Time Series
  - – Neural Forecasting Models
  - – Multivariate Time Series Analysis
  - – Anomaly Detection in Time Series

- **Autonomous Driving and Robotics**
  - – Perception for Autonomous Vehicles
  - – Path Planning and Control
  - – Simulation-to-Real Transfer
  - – Multi-Sensor Fusion
  - – Learning from Demonstration

- **Explainable Artificial Intelligence and Interpretability**
  - – Feature Attribution
  - – Counterfactual Explanations
  - – Concept-based Explanations
  - – Causal Interpretability

- **Adversarial Machine Learning**
  - – White-Box and Black-Box Attacks
  - – Adversarial Training
  - – Certified Robustness
  - – Physical Adversarial Attacks

- **Fairness, Accountability, and Transparency**
  - – Fairness Metrics
  - – Bias Mitigation Techniques
  - – Transparent Reporting
  - – Algorithmic Auditing

- **Privacy-Preserving Machine Learning**
  - – Differential Privacy
  - – Federated Privacy
  - – Homomorphic Encryption
  - – Secure Multi-Party Computation

- **Artificial Intelligence Safety and Reliability**

- Robustness Evaluation
- Out-of-Distribution Detection
- Alignment Research
- Verification and Formal Methods
- **Optimization for Machine Learning**
  - Stochastic Gradient Descent and Variants
  - Adaptive Optimizers
  - Second-Order Optimization Methods
  - Non-Convex Optimization
- **Model and System Efficiency**
  - Model Pruning
  - Quantization
  - Knowledge Distillation
  - Neural Architecture Search
  - Low-Rank Approximations
- **Causal Inference and Discovery**
  - Structural Causal Models
  - Causal Representation Learning
  - Invariant Risk Minimization
  - Counterfactual Reasoning
- **Artificial Intelligence for Science**
  - Molecular Simulation
  - Protein Folding
  - Climate Modeling
  - Material Science
- **Artificial Intelligence for Drug Discovery and Healthcare**
  - Drug-Target Interaction Prediction
  - Molecule Generation
  - Medical Image Analysis
  - Clinical Decision Support
- **Physics-Informed Machine Learning**
  - Physics-Informed Neural Networks
  - Differentiable Simulators
  - Scientific Machine Learning
- **Artificial Intelligence for Social Good**
  - Disaster Response
  - Public Health Modeling
  - Education and Accessibility
  - Environmental Monitoring

## A.3 PROMPTS

The LLM prompts for novelty evaluation in the Ego-Centric task are as follows:

```
You are an expert academic reviewer. Your task is to evaluate a
    scientific paper on its Novelty.

### Definition of Novelty:
Definition: Novelty refers to the uniqueness and originality of the
    research question, methodology, data, or conclusions relative
    to existing research.
Focus: Does the paper introduce new ideas, perspectives, or methods
    within the existing body of knowledge? For example, applying a
    method from Field A to Field B for the first time.
Scoring Criteria: A score of 0 represents completely derivative
    work, while a score of 10 represents a highly original and
    groundbreaking idea.

---

Please evaluate the Novelty of the following paper based on the
    definition provided.

**Title**: [Paper Title]

**Abstract**: [Paper Abstract]
(or: [No abstract provided. Please evaluate based on the title
    alone.])

---
```

```
Your response MUST be a single JSON object with 'score' (an integer
    from 0 to 10) and 'reasoning' (a brief explanation).
```

The LLM prompts for disruptiveness evaluation in the Ego-Centric task are as follows:

```
You are an expert academic reviewer. Your task is to evaluate a
    scientific paper on its Disruptiveness.

### Definition of Disruptiveness:
Definition: Disruptiveness refers to the way a paper influences
    subsequent research-does it cause future work to cite the paper
    itself, rather than the previous works it was built upon?
Focus: Does the paper change the direction of a research field or
    its methodologies, causing prior work to be marginalized? For
    example, the foundational papers on CRISPR gene-editing
    technology opened new research avenues and made previous
    editing methods obsolete.
Scoring Criteria: A score of 0 represents no disruptive potential
    (e.g., a review paper), while a score of 10 represents the
    potential to highly transform a field.

---

Please evaluate the Disruptiveness of the following paper based on
    the definition provided.

Title: [Paper Title]

Abstract: [Paper Abstract]
(or: [No abstract provided. Please evaluate based on the title
    alone.])

---

Your response MUST be a single JSON object with:
- "score": an integer from 0 to 10
- "reasoning": a brief explanation supporting the score
```

In the Pair-Wise task, the code for classifying the sentiment tendency of citation context using LLM is as follows:

```
You are an expert in academic literature analysis. Your task is to
    classify the sentiment of a citation context.

Please analyze the following citation context from a research
    paper, which mentions the target paper titled "[Target Paper
    Title]".

Classify the context into one of three categories:
- Positive: The citing paper praises, builds upon, or confirms the
    findings of the target paper.
- Negative: The citing paper criticizes, questions, or points out
    limitations of the target paper.
- Neutral: The citing paper simply mentions or describes the target
    paper as background information without expressing a strong
    opinion.
```

```
Your response MUST BE only ONE of the three category names:
    Positive, Negative, or Neutral.

Context to analyze:
---
[Insert citation context here]
---
```

The prompts for LLM evaluation in the Path-Wise task are as follows:

```
You are an expert in scientometrics and academic research. Your
    task is to evaluate the quality of a proposed citation path
    based on its technical evolution.

### Core Task:
Assess if the provided sequence of papers represents a logical and
    meaningful technological or conceptual evolution from the start
    paper to the end paper.

### What constitutes a good technical evolution path? (Key
    Principles)
1. Thematic Consistency: All papers must strictly revolve around
    the same core research topic defined by the query. Deviations
    into unrelated subjects indicate a poor path.
2. Content Cohesion & Logical Flow: The content of adjacent papers
    must be closely related. Each paper should logically follow
    from the previous one, building upon its ideas, refining its
    methods, or addressing its limitations.
3. Progressive Development: The path must demonstrate clear
    progress. Later papers should represent advancements,
    extensions, or significant new applications of the concepts
    introduced in earlier papers. The path should tell a story of
    innovation.
4. Represents a Main Line of Inquiry: The path should follow a
    significant and recognized line of development within the
    research field, not an obscure or tangential branch.

### Scoring Criteria (0-10):
- Score 9-10 (Excellent): A perfect or near-perfect path. It is
    thematically consistent, shows clear progressive development,
    and represents a major line of inquiry. The logical flow is
    impeccable.
- Score 7-8 (Good): A strong, coherent path. Most papers are
    relevant and show progression, but there might be a minor
    logical gap or a less influential paper included.
- Score 4-6 (Mediocre): The path has some relevance but lacks
    strong cohesion. It may include several tangential papers, the
    logical progression is weak, or it fails to capture the main
    developmental thread.
- Score 1-3 (Poor): The path is largely incoherent. Papers are
    thematically disconnected, show no clear progress, or are
    mostly irrelevant to the query.
- Score 0 (Failure): A completely random collection of papers with
    no logical or thematic connection.

---

Please evaluate the following citation path based on the detailed
    criteria provided.
```

```
Original Request: [original_query]

--- Proposed Citation Path ---
[Paper list will be inserted here]
------------------------------

Your response MUST be a single JSON object with 'score' (an integer
    from 0 to 10) and 'reasoning' (a detailed explanation for your
    score, critiquing the path based on the four key principles).
```

In the survey generation experiment 6, the prompts for generating the survey are as follows:

```
You are a helpful academic assistant that generates surveys using
    retrieved literature.

Please generate a survey report in Markdown format based on the
    following information:

Domain: [Domain Name]

Start paper:
Title: "[Start Paper Title]"
Abstract: [Start Paper Abstract]

End paper:
Title: "[End Paper Title]"
Abstract: [End Paper Abstract]

Task:
Generate a survey report describing the technological evolution
    from the start paper to the end paper.
Include key technical developments, major milestones, and method
    evolution.
Organize the report in a clear Markdown format.

Important:
- Do not use any ground-truth paths.
- Rely only on information retrieved via search capabilities.
```

