# OpenReview forum: "SciNetBench: A Relation-Aware Benchmark for Scientific Literature Retrieval Agents"
_ICLR.cc/2026/Conference — Submitted to ICLR 2026_

### Official Review · Reviewer_Yuen · 2025-10-26

**Soundness:** 3
**Presentation:** 2
**Contribution:** 3
**Rating:** 6
**Confidence:** 3

**Summary:**

This paper proposes SciNetBench for relation-aware literature retrieval agents, which is constructed from over 18 million AI papers and 1087 queries. It evaluates three levels of literature relations, including ego-centric, pair-wise, and path-wise retrieval. The experiments show that eight existing retrieval methods of three categories fail to capture these relations thus presenting new challenges in retrieval. To show the downstream application of the relation-aware retrieval, this work shows that the information of the relations helps agents improve the quality of literature review.

**Strengths:**

1. This paper proposes a benchmark for the relation-aware retrieval of three levels which helps a deeper understanding of literature.

2. The constructed benchmark includes 18 million AI papers that encompass diverse areas and evaluates existing retrieval methods from multiple aspects regarding the relation-aware retrieval.

3. The work also proposes a useful application of such relation-aware retrieval, that it helps improve the quality of the literature review.

**Weaknesses:**

1. The total corpus consists of over 18 million papers, which is comprehensive but might be costly in computation. Providing the details of the evaluation regarding the costs could be helpful for future work.

2. While the ego-centric and path-wise evaluation includes Scientometrics metrics or human evaluation, the pair-wise evaluation seems to only dependent on LLM evaluation from a few aspects. The LLM evaluation might prefer OpenAI models if other future deep research methods are compared.

3. The groundtruth for path-wise retrieval is generated by selecting the path with highest total citation, which might favor the path with more papers. It's also a concern that how much the citing between the papers on the path actually represents the evolution. For example, one paper cites another paper for 10 times might indicate more conceptually dependence on that paper.

4. Minor: the corpus include diverse tracks of AI research, while the information of the queries and tasks on their tracks or timespan is not included.

**Questions:**

In addition to the weaknesses listed above, how are the papers in the queries selected? Is there bias that famous or early-published papers are preferred in the queries?

---

### Official Review · Reviewer_iVTk · 2025-10-30

**Soundness:** 3
**Presentation:** 3
**Contribution:** 2
**Rating:** 6
**Confidence:** 3

**Summary:**

In this paper, the authors introduced SciNetBench, a benchmark for relation‑aware scientific literature retrieval across three granularities: (1) ego‑centric (estimating intrinsic properties such as novelty and disruptiveness), (2) pair‑wise (capturing relations between two papers, e.g., supportive/critical citation context and co‑mention in text), and (3) path‑wise (reconstructing multi‑step evolutionary trajectories between seminal and newer works). The authors built SciNetBench from an AI‑focused subset of OpenAlex (18.6M papers) and arXiv PDFs. It defines task‑specific protocols and evaluates eight systems including embedding‑based retrievers, agentic/web‑search systems, and deep research-style agents. Results show low performance on relation‑aware tasks (for example, novelty Recall@50 <= 1.42% and disruption Recall@50 <= 4.60%), with deep research and web search agents showing strongest results while still far from being satisfying on sentiment and paragraph‑level co‑mentions and on constructing coherent paths. The authors also demonstrated that providing ground‑truth relational signals improves downstream survey generation quality.

**Strengths:**

- The authors did not rely on the more common topical matching, and extended it to network‑aware retrieval, which is important for actual research workflows.
- The three task granularities that the authors proposed are well‑motivated, technically sound, and complementary to each other.
- The authors adopted robust metrics such as Uzzi novelty, Funk–Owen‑Smith disruption, and credit allocation.
- SciNetBench is a large-scale, open-source benchmark, which will be very valuable for the community.
- The evaluation is comprehensive, with extensive baselines, thorough analysis, and useful downstream case study.

**Weaknesses:**

- Ground‑truth path heuristic may bias to popularity. Selecting the path with maximum cumulative citations introduced the risk of preferencing older/highly cited detours over the widely accepted or methodologically coherent trajectories.
- Novelty‑LLM / Disruption‑LLM and Rationality‑LLM rely on a single model. The authors should consider multi‑judge aggregation or report inter‑rater reliability.
- Some strongest baselines (web‑search and deep‑research agents) are proprietary API-based and change over time. This would cause issues of the reproducibility of the paper.

**Questions:**

- What maximum path length is allowed in BFS?

- Have the authors tried multiple LLM judges or adjudication? Any agreement stats between LLM and human ratings (beyond the path‑wise human rationality subset)?

---

### Official Review · Reviewer_ZRkQ · 2025-10-31

**Soundness:** 2
**Presentation:** 3
**Contribution:** 2
**Rating:** 2
**Confidence:** 4

**Summary:**

This paper introduces a new benchmark for scientific literature retrieval agents, designed to evaluate their ability to perform relation-aware reasoning on citation networks. The proposed benchmark comprises three types of queries: ego-centric relation queries, pair-wise relation queries, and path-wise relation queries. The authors further examine several existing models, including SciBERT, PaperQA2, and DeepSearch on this dataset, revealing their limitations in handling such relation-aware retrieval tasks.

**Strengths:**

1. This paper is well-written and clearly structured, making it easy to follow. The figures are well designed and effectively facilitate the understanding of both dataset construction and ground-truth labels.

2. Retrieving scientific literature with relation-aware conditions is a highly interesting research task. I believe this study holds significant potential for real-world applications.

**Weaknesses:**

1. The scope of the proposed benchmark dataset is limited, as it includes only AI-related papers and therefore lacks the generality needed to thoroughly evaluate relation-aware retrieval capabilities across diverse scenarios.

2. Although a key novelty of this benchmark lies in its construction of three types of queries, the total number of queries remains relatively small (only 1,000), which hinders effective model training and comprehensive evaluation. Moreover, it is unclear why the authors opted to extract a new citation graph, given the availability of established citation graphs such as PubMed and OGB-arxiv, which already contain relevant graph structures. These existing datasets could also readily support the construction of the proposed queries. The ground-truth construction of these queries is unconvincing. In ego-centric relation queries, the authors' heuristic that "novel" papers must be those that bridge different research topics is not necessarily valid. A paper can demonstrate novelty within a single, well-established domain without significant cross-topic integration. Conversely, a paper that bridges topics might simply apply established methods from one area to another without introducing fundamental novelty. This heuristic may introduce noise and bias into the ground-truth labels. In path-wise relation queries, the assumption that the highest citation number path in the citation graph represents the genuine evolution of a technical idea is problematic. The actual lineage of ideas is often more complex and cannot be reliably reduced to a max citation path. This construction method cannot guarantee that the identified path truly reflects the historical and intellectual flow of technological development, as citations can be made for various reasons (e.g., background, comparison) rather than direct lineage.

**Questions:**

See above comments.

---

### Official Review · Reviewer_JK5g · 2025-11-02

**Soundness:** 2
**Presentation:** 3
**Contribution:** 2
**Rating:** 2
**Confidence:** 5

**Summary:**

This paper introduces SciNetBench (Scientific Network Relation-aware Benchmark for literature retrieval agents), which is a benchmark built from a very large AI-literature corpus, to evaluate whether retrieval systems and research ``agents'' can go beyond topical similarity and reason over relations in a scientific network. It proposes three tasks: 1. ego-centric (novelty/disruption of a single paper); 2. pair-wise (sentiment of a citation, co-mention in context); and 3. path-wise (reconstructing citation paths/lineages), and evaluates eight systems spanning embedding baselines, search-augmented agents, and ``deep research'' agents. Experimental results show that current systems often perform poorly (e.g., <20% on key performance metrics), but relational ground truth can help improve models' performance. Experiments are also performed on a downstream literature-review task.

**Strengths:**

1. The paper introduces a large benchmark, SciNetBench, built on top of OpenAlex (full snapshot) and contains 18M+ AI papers across 177 subfields.

2. Taking into account the relations between papers through citations contexts is good although not a novel idea.

3. The baselines used in the paper cover embedding-based, agentic search, and deep research agents.

**Weaknesses:**

1. Although the dataset is interesting, I find the novelty of this paper limited. I also wonder if the components that are added to the system overall do not in fact introduce biases. For example, first, for the novelty of the paper, besides putting together some metrics from the literature (e.g., novelty by Uzzi et al or disruption by Funk and Owen-Smith), I do not see the novelty in the creation of this new benchmark. Second, for the novelty / disruption, the question that is used in the paper: "What are the most novel papers in diffusion language models?" may introduce a lot of biases (besides some seminal papers, this becomes a subjective measure and I worry about the evaluation of this knowledge structure component). Similarly for the sentiment captured through peer assessment.

2. The data annotation for the evaluation process is not fully detailed, and hence, it is unclear what the retrieval quality is. For a new benchmark I would expect to see full details of the annotation process. For the human evaluation, who are the annotators? What qualifications they have? How comprehensively did they go through the results/annotations?

3. An error analysis is missing from the paper and would strengthen its quality.

**Questions:**

How to address the biases introduced through the approach for each one of the components of the system?

How was the human evaluation performed? Details are missing.

How does this approach compare with commercialized systems for scientific retrieval?

What kinds of errors do the models make?

---

### Meta-Review · Area_Chair_WCuq · 2026-01-07

**Summary:**

This paper proposes SciNetBench to evaluate relation-aware scientific retrieval, moving beyond simple topical similarity. While all reviewers appreciate the motivation and the large-scale corpus (18M+ papers), the execution suffers from fundamental validity issues. The primary concerns stem from the heuristics used to generate ground-truth labels—specifically, defining "novelty" via topic bridging and "evolutionary paths" via citation counts—which reviewers argue introduce significant noise and popularity bias. Additionally, the restriction to a single domain (AI) and the lack of transparency regarding human evaluation protocols limit the benchmark's utility.

**Reviewer Concerns:**

No rebuttal has been made

**Reviewer Scores:**

No rebuttal has been made, therefore scores won't change

---

### Decision · Program_Chairs · 2026-01-26

Reject